# The Effect of Cadmium on GFR Is Clarified by Normalization of Excretion Rates to Creatinine Clearance

**DOI:** 10.3390/ijms22041762

**Published:** 2021-02-10

**Authors:** Soisungwan Satarug, David A. Vesey, Muneko Nishijo, Werawan Ruangyuttikarn, Glenda C. Gobe, Kenneth R. Phelps

**Affiliations:** 1Kidney Disease Research Collaborative, Translational Research Institute, The University of Queensland, Brisbane 4102, Australia; David.Vesey@health.qld.gov.au (D.A.V.); g.gobe@uq.edu.au (G.C.G.); 2Department of Nephrology, Princess Alexandra Hospital, Woolloongabba, Brisbane 4075, Australia; 3Department of Public Health, Kanazawa Medical University, Uchinada, Ishikawa 920-0293, Japan; ni-koei@kanazawa-med.ac.jp; 4Division of Toxicology, Faculty of Medicine, Chiang Mai University, Chiang Mai 50200, Thailand; ruangyuttikarn@gmail.com; 5School of Biomedical Sciences, The University of Queensland, Brisbane 4072, Australia; 6NHMRC Centre of Research Excellence for CKD, QLD, UQ Health Sciences, Royal Brisbane and Women’s Hospital, Brisbane 4029, Australia; 7Stratton Veterans Affairs Medical Center, Albany, New York 12208, USA; 8Albany Medical College, Albany, New York 12208, USA

**Keywords:** cadmium, chronic kidney disease, excretion rate, GFR, N-acetyl-β-D-glucosaminidase, nephrotoxicity, urine creatinine

## Abstract

Erroneous conclusions may result from normalization of urine cadmium and N-acetyl-β-D-glucosaminidase concentrations ([Cd]_u_ and [NAG]_u_) to the urine creatinine concentration ([cr]_u_). In theory, the sources of these errors are nullified by normalization of excretion rates (E_Cd_ and E_NAG_) to creatinine clearance (C_cr_). We hypothesized that this alternate approach would clarify the contribution of Cd-induced tubular injury to nephron loss. We studied 931 Thai subjects with a wide range of environmental Cd exposure. For *x* = Cd or NAG, E*_x_*/E_cr_ and E*_x_*/C_cr_ were calculated as [*x*]_u_/[cr]_u_ and [*x*]_u_[cr]_p_/[cr]_u_, respectively. Glomerular filtration rate (GFR) was estimated according to the Chronic Kidney Disease (CKD) Epidemiology Collaboration (eGFR), and CKD was defined as eGFR < 60 mL/min/1.73m^2^. In multivariable logistic regression analyses, prevalence odds ratios (PORs) for CKD were higher for log(E_Cd_/C_cr_) and log(E_NAG_/C_cr_) than for log(E_Cd_/E_cr_) and log(E_NAG_/E_cr_). Doubling of E_Cd_/C_cr_ and E_NAG_/C_cr_ increased POR by 132% and 168%; doubling of E_Cd_/E_cr_ and E_NAG_/E_cr_ increased POR by 64% and 54%. As log(E_Cd_/C_cr_) rose, associations of eGFR with log(E_Cd_/C_cr_) and log(E_NAG_/C_cr_) became stronger, while associations of eGFR with log(E_Cd_/E_cr_) and log(E_NAG_/E_cr_) became insignificant. In univariate regressions of eGFR on each of these logarithmic variables, R^2^ was consistently higher with normalization to C_cr_. Our tabular and graphic analyses uniformly indicate that normalization to C_cr_ clarified relationships of E_Cd_ and E_NAG_ to eGFR.

## 1. Introduction

Cadmium (Cd) is an important industrial toxin in several regions of the world [1]. It enters the human body in food, cigarette smoke, and polluted air, and is initially bound to the protein metallothionein (MT) in the liver. Complexes of CdMT are subsequently released to the circulation, filtered by renal glomeruli, and reabsorbed by proximal tubules. After Cd is separated from MT in lysosomes of tubular cells, it induces in situ synthesis of MT, which mitigates the toxicity of the free metal. Nevertheless, a small amount of unbound Cd inflicts injury that may eventuate in nephron loss and a reduction in the glomerular filtration rate (GFR) [2,3].

The lysosomal enzyme N-acetyl-β-D-glucosaminidase (NAG) is the most commonly employed marker of Cd-induced cell injury. Because this molecule is too large to undergo glomerular filtration, its appearance in urine signifies release by injured tubular cells [4]. The excretion rate of NAG (E_NAG_) typically correlates with that of Cd (E_Cd_) [5,6,7,8,9,10,11], and the correlation holds at minimal Cd excretion rates [12,13,14]. Given these observations, we have argued that the two substances probably emanate from the same source [15]. We recently showed that E_Cd_ and E_NAG_ were inversely related to estimated glomerular filtration rate (eGFR) in a sample of Thai subjects [15].

Historically, investigators of Cd nephrotoxicity have normalized the excretion rates of various substances (E*_x_*) to that of creatinine (E_cr_) [7,9,10,11,12,16,17]. Algebraically, E*_x_*/E_cr_ simplifies to [*x*]_u_/[cr]_u_. Although normalization of [*x*]_u_ to [cr]_u_ corrects for the effect of urine flow rate (V_u_) on [*x*]_u_, this convention introduces two other sources of error. First, because E_cr_ is primarily a function of muscle mass [18], [*x*]_u_/[cr]_u_ may vary by a multiple among subjects with a given E*_x_*. Second, if Cd and NAG emanate from tubular cells, their excretion rates may fall as nephrons are lost while E_cr_ remains relatively stable [19,20,21]. In this circumstance, E_Cd_/E_cr_ and E_NAG_/E_cr_ may understate the severity of Cd-induced injury.

To avoid these potential errors, we recently adopted the practice of normalizing E*_x_* to creatinine clearance (C_cr_), a surrogate for the glomerular filtration rate (GFR) [15,22]. C_cr_ is the excretion rate divided by the plasma concentration of creatinine (E_cr_/[cr]_p_); algebraically, E*_x_*/C_cr_ simplifies to [*x*]_u_[cr]_p_/[cr]_u_ in units of mass of *x* excreted per volume of filtrate [23]. Because the ratio [*x*]_u_/[cr]_u_ remains in the calculation, E*_x_*/C_cr_, like E*_x_*/E_cr_, is unaffected by V_u_. However, in contrast to E*_x_*/E_cr_, E*_x_*/C_cr_ is also unaffected by muscle mass because [cr]_p_ varies in proportion to E_cr_ at a given C_cr_. Moreover, if *x* emanates from tubular cells, E*_x_* may fall because of nephron loss; in that circumstance, E*_x_*/C_cr_ depicts the excretion of *x* per surviving nephron because C_cr_ falls simultaneously [2,3].

Given the foregoing considerations, we hypothesized that normalization of E_Cd_ and E_NAG_ to C_cr_ would clarify the contribution of Cd-induced tubular injury to nephron loss. We employed data from a large sample of Thai subjects to examine regressions of estimated GFR (eGFR) on log(E_Cd_/E_cr_), log(E_NAG_/E_cr_), log(E_Cd_/C_cr_), and log(E_NAG_/C_cr_). Normalization to C_cr_ increased coefficients of determination, effect sizes, and the strength of associations of eGFR with excretion rates.

## 2. Results

### 2.1. Tabular Analyses

Table 1 summarizes demographic features, renal function, and excretion rates of Cd and NAG in a cohort of 931 subjects. Data were organized in three subsets defined by gradations of log[(E_Cd_/C_cr_) × 10^5^]. As log(E_Cd_/C_cr_) rose, age, [Cd]_u_, [NAG]_u_, E_Cd_/E_cr_, E_NAG_/E_cr_, E_Cd_/C_cr_, and E_NAG_/C_cr_ also rose, and eGFR fell.

Table 2 presents two logistic regression models that quantified contributions of independent variables to the prevalence odds ratio (POR) for CKD (eGFR < 60 mL/min/1.73m^2^). Both models incorporated demographic factors and parameters of Cd and NAG excretion. E_Cd_ and E_NAG_ were normalized to E_cr_ in model 1 and to C_cr_ in model 2.

In model 1, Table 2, POR for CKD was associated with age, log_2_(E_Cd_/E_cr_), and log_2_(E_NAG_/E_cr_), but not with diabetes, gender, hypertension, or smoking. Log_2_(E_Cd_/E_cr_) had the greatest effect size (β = 0.493), followed by log_2_(E_NAG_/E_cr_) (β = 0.435) and age (β = 0.143). In model 2, results were qualitatively similar; POR was associated with age, log_2_(E_Cd_/C_cr_), and log_2_(E_NAG_/C_cr_), but not with diabetes, gender, hypertension, or smoking. In contrast to model 1, log_2_(E_NAG_/C_cr_ had the greatest effect size (β = 0.985), followed by log_2_E_Cd_/C_cr_ (β = 0.843) and age (β = 0.149). Effect size on POR was much greater for log_2_(E_Cd_/C_cr_) and log_2_(E_NAG_/C_cr_) than for log_2_(E_Cd_/E_cr_) and log_2_(E_NAG_/E_cr_). Doubling of E_Cd_/E_cr_ and E_NAG_/E_cr_ increased POR for CKD by 63.7% and 54.4%, respectively; doubling of E_Cd_/C_cr_ and E_NAG_/C_cr_ increased POR by 132% and 168%, respectively. In both models, POR for CKD rose by 14–15% with every 10-year increment above age 40.

Table 3 presents two multivariable linear regression models of eGFR. As in Table 2, the models incorporated demographic factors and parameters of Cd and NAG excretion. E_Cd_ and E_NAG_ were normalized to E_cr_ in model 1 and to C_cr_ in model 2. Accordingly, subsets of subjects were defined by gradations of log[(E_Cd_/E_cr_) × 10^3^] in model 1 and log[(E_Cd_/C_cr_) × 10^5^] in model 2. In each model, age exhibited the greatest absolute value of standardized β (strength of association) at all three gradations of log[(E_Cd_/E_cr_) × 10^3^] or log[(E_Cd_/C_cr_) × 10^5^].

In model 1, Table 3, other demographic variables were also associated with eGFR, but only in the subset with the lowest log(E_Cd_/E_cr_). As log(E_Cd_/E_cr_) rose, standardized β became more negative and its absolute value increased for the regression of eGFR on log(E_Cd_/E_cr_). In contrast, standardized β remained positive for the regression of eGFR on log(E_NAG_/E_cr_). In the subset with the highest log(E_Cd_/E_cr_), neither of the regressions reached statistical significance. Variation in eGFR accounted for by all independent variables (adjusted R^2^) *fell* as log(E_Cd_/E_cr_) rose and eGFR fell.

In model 2, Table 3, age was the only demographic factor associated with eGFR. As log(E_Cd_/C_cr_) rose, standardized β became more negative and its absolute value increased for regressions of eGFR on log(E_Cd_/C_cr_) and log(E_NAG_/C_cr_). Both regressions were highly significant in the middle and highest subsets of log(E_Cd_/C_cr_). Variation in eGFR accounted for by all independent variables (adjusted R^2^) *rose* as log(E_Cd_/C_cr_) rose and GFR fell.

### 2.2. Graphic Analyses

Figure 1 compares linear and quadratic regressions of eGFR on log(E_Cd_/E_cr_) (graph (**A**)) and log(E_Cd_/C_cr_) (graph (**C**)). The graphs show that R^2^ was higher and *p* was lower for regressions of eGFR on log(E_Cd_/C_cr_). Tables (**B**) and (**D**) enumerate R^2^ and unstandardized and standardized β for linear regressions in subsets defined by increasing lower limits of log(E_Cd_/E_cr_) or log(E_Cd_/C_cr_). In all subsets and in the sample as a whole, R^2^ and absolute values of unstandardized and standardized β were higher for regressions of eGFR on log(E_Cd_/C_cr_).

Figure 1E depicts the mean eGFR of women and men in each of three subsets defined by ranges of log(E_Cd_/E_cr_). The ranges are those employed to denote levels of Cd excretion in Table 3, model 1. After adjustment for covariates and interactions, a significant difference in mean eGFR was demonstrated between women in the highest and lowest subsets. No differences were found among men. Figure 1F depicts the mean eGFR of women and men in each of three subsets defined by ranges of log(E_Cd_/C_cr_). The ranges are those employed to denote levels of Cd excretion in Table 3, model 2. After adjustment for covariates and interactions, significant differences in mean eGFR were demonstrated between both women and men in the lowest subset and their counterparts in the middle and highest subsets.

Figure 2 compares linear and quadratic regressions of eGFR on log(E_NAG_/E_cr_) (graph (**A**)) and log(E_NAG_/C_cr_) (graph (**C**)). The graphs show that R^2^ was higher and *p* was lower for regressions of eGFR on log(E_NAG_/C_cr_). Tables (**B**) and (**D**) enumerate R^2^ and unstandardized and standardized β for linear regressions in subsets defined by increasing lower limits of log(E_NAG_/E_cr_) or log(E_NAG_/C_cr_). In all subsets and in the sample as a whole, R^2^ and absolute values of unstandardized and standardized β were higher for regressions of eGFR on log(E_NAG_/C_cr_). In two subsets, only the regressions of eGFR on log(E_NAG_/C_cr_) were significant.

Figure 2E depicts the mean eGFR of women and men in each of three subsets defined by ranges of log(E_NAG_/E_cr_). After adjustment for covariates and interactions, a significant difference was demonstrated between men in the highest and lowest subsets. No differences in eGFR were seen among women. Figure 2F depicts the mean eGFR of women and men in each of three subsets defined by log(E_NAG_/C_cr_). After adjustment for covariates and interactions, significant differences were demonstrated between both women and men in the lowest subset and their counterparts in the middle and highest subsets.

Figure 3 compares linear and quadratic regressions of log(E_NAG_/E_cr_) on log(E_Cd_/E_cr_) (graph (**A**)) or log(E_NAG_/C_cr_) on log(E_Cd_/C_cr_) (graph (**C**)). The graphs show that R^2^ was higher and *p* was lower for regressions of log(E_NAG_/C_cr_) on log(E_Cd_/C_cr_). Tables (**B**) and (**D**) examine R^2^ and unstandardized and standardized β in subsets defined by progressively increasing lower limits of log(E_Cd_/E_cr_) or log(E_Cd_/C_cr_). In all but one of the subsets and in the sample as a whole, R^2^ and the absolute values of unstandardized and standardized β were higher for regressions of log(E_NAG_/C_cr_) on log(E_Cd_/C_cr_). Regressions of log(E_NAG_/C_cr_) on log(E_Cd_/C_cr_) were significant in three of the subsets, but regressions of log(E_NAG_/E_cr_) on log(E_Cd_/E_cr_) were significant in only one.

Figure 3E depicts log(E_NAG_/C_cr_) of women and men in each of three subsets defined by ranges of log(E_Cd_/E_cr_). After adjustment for covariates and interactions, a significant difference in mean log(E_NAG_/E_cr_) was demonstrated between women in the highest and lowest subsets, and between men in the lowest and the middle and highest subsets. Figure 3F depicts log(E_NAG_/C_cr_) of women and men in each of three subsets defined by ranges of log(E_Cd_/C_cr_). After adjustment for covariates and interactions, significant differences in mean log(E_NAG_/C_cr_) were demonstrated between both women and men in the lowest subset and their counterparts in the middle and highest subsets.

## 3. Discussions

We have previously argued that regressions of eGFR on parameters of Cd and NAG excretion provide insight into the pathogenesis of Cd nephropathy [15]. Heretofore, investigators of this issue have normalized [Cd]_u_ and [NAG]_u_ to [cr]_u_ [7,8,16,17]. The resulting ratios adjust [Cd]_u_ and [NAG]_u_ for variation in V_u_, but muscle mass affects [cr]_u_, and nephron number affects excretion rates of substances that emanate from tubular cells. In theory, normalization of E_Cd_ and E_NAG_ to C_cr_ resolves these issues for reasons summarized in the Introduction. We therefore hypothesized that this methodological modification would add clarity to relationships among eGFR, E_Cd_, and E_NAG_.

### 3.1. Interpretation of Tabular Analyses

We created the study sample to encompass a broad range of probability that asymptomatic participants had developed Cd nephropathy. To highlight the relevance of previous Cd accumulation to GFR, we divided the sample into three subsets defined by gradations of log(E_Cd_/C_cr_). As log(E_Cd_/C_cr_) rose, age, E_Cd_/E_cr_, E_NAG_/E_cr_, E_Cd_/C_cr_, and E_NAG_/C_cr_ also rose, and eGFR fell (Table 1).

We performed two multivariable logistic regression analyses to quantify effects of demographic features and excretory parameters on the probability of CKD (eGFR < 60 mL/min/1.73m^2^) (Table 2). The unit of probability was the ratio of odds of having CKD to odds of not having it, i.e., the probability odds ratio (POR). POR was associated with log_2_(E_Cd_/E_cr_) and log_2_(E_NAG_/E_cr_) in model 1 and with log_2_(E_Cd_/C_cr_) and log_2_(E_NAG_/C_cr_) in model 2. POR was higher for logs of the ratios in model 2, as were the percentage increases in POR per doubling of ratios.

We also performed two regression model analyses of eGFR on demographic variables and parameters of Cd and NAG excretion (Table 3). Each regression was examined in three subsets defined by progressively higher ranges of log(E_Cd_/E_cr_) (model 1) or log(E_Cd_/C_cr_) (model 2). The ranges were designated Cd excretion levels 1, 2, and 3. In model 1, associations of eGFR with log(E_Cd_/E_cr_) and log(E_NAG_/E_cr_) became *less* significant as log(E_Cd_/E_cr_) rose in the subsets. Moreover, at excretion level 3, standardized β for the regression of eGFR on E_NAG_/E_cr_ was positive rather than negative, and thus implied the absence of an inverse relationship between eGFR and the severity of tubular injury. In model 2, as log(E_Cd_/C_cr_) rose in the Cd excretion subsets, associations of eGFR with E_Cd_/C_cr_ and E_NAG_/C_cr_ became *more* significant, the absolute value of standardized β (strength of association) increased, the slope implied by standardized β became more negative, the inverse relationship of eGFR to tubular injury was thus enhanced, and R^2^ for the entire analysis rose.

Collectively, our tabular data demonstrate multiple benefits of normalizing excretion rates to C_cr_ rather than E_cr_. This approach substantially magnified the effects of E_Cd_ and E_NAG_ on POR for CKD; exposed the inverse relationship between eGFR and E_NAG_ at high levels of Cd excretion; demonstrated qualitatively similar regressions of eGFR on log(E_Cd_/C_cr_) and log(E_NAG_/C_cr_); increased the size of effects of E_Cd_ and E_NAG_ on eGFR and the strength of associations among these variables; and produced a multilinear regression model that accounted for more variation in eGFR.

### 3.2. Interpretation of Graphic Analyses

The univariate analyses in Figure 1 and Figure 2 confirm the concepts imparted by Table 3. Figure 1 depicts linear and quadratic regressions of eGFR on log(E_Cd_/E_cr_) or log(E_Cd_/C_cr_). R^2^ was higher and *p* was lower for both types of regression when eGFR was plotted against log(E_Cd_/C_cr_); moreover, dispersion was visibly reduced and a curvilinear relationship between eGFR and E_Cd_ was more evident (Figure 1A,C). Slope analyses were performed over ranges of log(E_Cd_/E_cr_) or log(E_Cd_/C_cr_) that were progressively reduced by raising lower limits. Within each range, effect size (unstandardized β) and strength of association (standardized β) were greater for relationships of eGFR to log(E_Cd_/C_cr_). Subsets created in Table 3 according to Cd excretion level were employed in Figure 1E,F, and differences in mean eGFR were more pronounced among subsets defined by log(E_Cd_/C_cr_).

Figure 2 depicts linear and quadratic regressions of eGFR on log(E_NAG_/E_cr_) or log(E_NAG_/C_cr_). R^2^ was much higher for both regressions when eGFR was plotted against log(E_NAG_/C_cr_); simultaneously, dispersion was reduced, and both linear and quadratic relationships of eGFR to E_NAG_ were more visually evident. As in Figure 1, slope analyses were performed over ranges of log(E_Cd_/E_cr_) or log(E_Cd_/C_cr_) that were progressively reduced by raising lower limits. Within each range, effect size (unstandardized β) and strength of association (standardized β) were much greater for relationships of eGFR to log(E_NAG_/C_cr_). Linear regressions relevant to each range were significant only for the highest range of E_NAG_/E_cr_, but they were highly significant for all but the lowest range of E_NAG_/C_cr_. Three subsets were created within the sample according to low, medium, or high ranges of log(E_NAG_/E_cr_) or log(E_NAG_/C_cr_); as in Figure 1, differences in mean eGFR were more pronounced among subsets defined by log(E_NAG_/C_cr_).

Figure 3 depicts linear and quadratic regressions of log(E_NAG_/E_cr_) on log(E_Cd_/E_cr_), and log(E_NAG_/C_cr_) on log(E_Cd_/C_cr_). Although the graphs are visually similar, a reduction in dispersion was demonstrated in the plot of E_NAG_/C_cr_ against E_Cd_/C_cr_, and R^2^ was higher for both linear and quadratic regressions of log(E_NAG_/C_cr_) on log(E_Cd_/C_cr_). In the slope analyses, R^2^ and unstandardized and standardized β and were higher for each individual linear regression of log(E_NAG_/C_cr_) on log(E_Cd_/C_cr_). As in Figure 1, subsets created in Table 3 according to Cd excretion level were employed in Figure 3E,F. Differences in NAG excretion were more pronounced when subsets were defined by log(E_Cd_/C_cr_). Thus, in all three figures, normalization of excretion rates to C_cr_ rather than E_cr_ increased coefficients of determination, effect size, and strength of association for each possible bivariate relationship among eGFR, E_Cd_, and E_NAG_. Within each gender, normalization to C_cr_ also accentuated differences in eGFR and NAG excretion among subsets defined by Cd excretion.

### 3.3. Creatinine Excretion, Creatinine Clearance, and GFR

Creatinine, a small nitrogenous waste product (m.w. 113 Da), is synthesized from creatine phosphate in skeletal muscle. It can enter plasma from the gut if meat is ingested, but most of its flux into plasma results from endogenous production in muscle cells. A small fraction of the influx is diverted to the colonic lumen for bacterial metabolism, and that fraction increases as [cr]_p_ rises. Nevertheless, at all but the most severely reduced values of GFR, renal excretion is the principal avenue of creatinine elimination [25,26].

When plasma is in equilibrium with respect to creatinine, the principal determinant of E_cr_ is muscle mass, which is highly variable in the population [18]. Most excreted creatinine is filtered, and the remainder is secreted by proximal tubules. The secreted fraction is small when GFR is normal; however, as GFR falls, the secreted fraction rises, C_cr_ (=E_cr_/[cr]_p_) increasingly overestimates GFR [26,27], and in theory, E_Cd_/C_cr_ and E_NAG_/C_cr_ increasingly underestimate the excretion of Cd and NAG per volume of filtrate. If C_cr_ were more uniformly representative of GFR, the relationships of eGFR to log(E_Cd_/C_cr_) and log(E_NAG_/C_cr_) would likely be less quadratic and more linear than Figure 1 and Figure 2 indicate. In any case, the inverse nature of these relationships is indisputable.

Estimated GFR approximates the radionuclide-based gold standard of GFR determination more closely than C_cr_ does [24]. One could argue, therefore, that we should normalize E_Cd_ and E_NAG_ to eGFR in our work. In theory, this method would offer the same advantages as normalization to C_cr_, and at low GFR, it would prevent underestimation of E_Cd_ and E_NAG_ per volume of filtrate. The practical obstacle to this approach is that determinations of E_Cd_/eGFR and E_NAG_/eGFR would require timed urine collections for the measurement of E_Cd_ and E_NAG_. Normalization to C_cr_, though possibly less accurate, is more convenient and less susceptible to procedural error because it is accomplished with single aliquots of serum and urine.

### 3.4. Tubular Release of Cd and NAG Necessitates Normalization of Excretion Rates to C_cr_

Excreted NAG emanates exclusively from injured tubular cells because the molecule is too large to undergo glomerular filtration [4]. The source of excreted Cd is more debatable, but several considerations suggest that Cd is also released from injured cells [15]. In multiple studies, E_Cd_ varied directly, not inversely, with GFR (nephron number) [19,20,21]. Similarly, E_Cd_ varied directly with the Cd content of kidneys sampled at autopsy or transplantation [28,29]. Because animal studies indicated that the tubular reabsorptive capacity for filtered Cd is quite high [30], it is unlikely that a typically intoxicated human excretes unreabsorbed Cd immediately after filtration. Previously reported correlations of E_Cd_ with E_NAG_ suggest that Cd and NAG emanate from a common source [5,6,7,8,9,10,11,12,13,14,15]; consequently, we have argued that both Cd and NAG are released into glomerular filtrate from tubular cells. If this inference is correct, then E_Cd_, like E_NAG_, is an indicator of Cd-induced tubular injury [15]. Tubulointerstitial nephritis, destruction of nephrons, and a reduction in GFR are logical sequelae of such injury [3,31].

If urinary Cd and NAG emanate from tubules, then we should expect the excretion of these substances to vary directly with the number of intact nephrons and the severity of cellular injury. In the absence of renal hypoperfusion, we assume a proportional relationship between GFR and nephron number [3]. Consequently, to focus on the severity of injury as a determinant of eGFR, we nullify the simultaneous contribution of nephron number to E_Cd_ and E_NAG_ by normalizing these excretion rates to C_cr_. Whereas the relationship of E_cr_ to nephron number is indirect and highly variable among subjects, the relationship of C_cr_ to nephron number is relatively direct and consistent. On theoretical grounds, we expect eGFR to be more closely associated with E_Cd_/C_cr_ and E_NAG_/C_cr_ than with E_Cd_/E_cr_ and E_NAG_/E_cr_, and the data reported herein confirm that expectation. We recommend the adoption of C_cr_ as the optimal denominator for the normalization of excretion rates in studies of Cd nephropathy.

## 4. Materials and Methods

### 4.1. Study Population

To develop a diverse sample with a wide range of environmental exposure to Cd, we assembled archived data drawn from multiple sites in Thailand. At the time of recruitment, all participants had lived at their current addresses for at least 30 years, and all gave informed consent to participate. Exclusion criteria were pregnancy, breast-feeding, a history of metal work, and a hospital record or physician’s diagnosis of an advanced chronic disease. Smoking, diabetes, hypertension, regular use of medications, educational level, occupation, and family health history were ascertained by questionnaire. Diabetes was defined as fasting plasma glucose levels ≥126 mg/dL or a physician’s prescription of anti-diabetic medications. Hypertension was defined as systolic blood pressure ≥140 mmHg, diastolic blood pressure ≥90 mmHg, a physician’s diagnosis, or prescription of anti-hypertensive medications.

Blood and urine were obtained in 2001 and 2002 from control subjects in Bangkok, and in 2004 and 2005 from subjects in subsistence farming areas of Mae Sot District. As judged by the Cd content of rice, a dietary staple in Thailand, exposure to Cd was low in Bangkok and moderate or high in Mae Sot [17,32,33,34]. Because occupational exposure was an exclusion criterion, we presumed that all participants had acquired Cd from the environment. After exclusion of participants with incomplete datasets, the study sample included 545 women and 386 men. Within the sample, urinary Cd varied by a factor >1000; age ranged from 16 to 87 years, and eGFR from 20 to 139 mL/min/1.73 m^2^. The Institutional Ethical Committees of Chulalongkorn University, Chiang Mai University, and the Mae Sot Hospital approved the study protocol [33].

### 4.2. Specimen Collection and Analysis

Second morning-void urine samples were collected after an overnight fast. Within the ensuing 3 h, specimens of whole blood were obtained and serum samples were prepared. Aliquots of urine, whole blood, and serum were transported on ice from a mobile clinic to a laboratory and stored at −20 or −80 °C for later analysis. Assays of creatinine in urine and serum ([cr]_u_, [cr]_p_]) were based on the Jaffe reaction. The assay of NAG in urine ([NAG]_u_) was based on colorimetry (NAG test kit, Shionogi Pharmaceuticals, Sapporo, Japan).

For the Bangkok group, [Cd]_u_ was determined by inductively-coupled plasma mass spectrometry (ICP/MS, Agilent 7500, Agilent Technologies, Santa Clara, CA, USA) because this method was sufficiently sensitive to measure Cd concentrations below the detectable limit of atomic absorption spectrophotometry. Multi-element standards (EM Science, EM Industries, Inc., Newark, NJ, USA) were used to calibrate Cd analyses, and accuracy and precision of those analyses were evaluated with reference urine (Lyphochek^®^, Bio-Rad, Gladesville, New South Wales, Australia). When [Cd]_u_ was less than the detection limit, 0.05 μg/L, the concentration assigned was the detection limit divided by the square root of 2. Fifty-eight subjects (14.7%) in the Bangkok group had [Cd]_u_ < 0.05 μg/L.

For the Mae Sot groups, [Cd]_u_ was determined by atomic absorption spectrophotometry (Shimadzu Model AA-6300, Kyoto, Japan). Urine standard reference material No. 2670 (National Institute of Standards, Washington, DC, USA) was used for quality assurance and control purposes. None of the urine samples from the Mae Sot groups were found to have [Cd]_u_ below the detection limit.

### 4.3. Normalization of Excretion Rates to E_cr_ and C_cr_

Excretion rates of Cd and NAG (E_Cd_ and E_NAG_) were normalized to E_cr_ or to C_cr_, a surrogate for GFR. E_Cd_/E_cr_ and E_NAG_/E_cr_ were expressed as mass excreted per g of creatinine; E_Cd_/C_cr_ and E_NAG_/C_cr_ were expressed as mass excreted per volume of filtrate. For *x* = Cd or NAG, E_x_/E_cr_ was calculated as [*x*]_u_/[cr]_u_, and E*_x_*/C_cr_ was calculated as [*x*]_u_[cr]_p_/[cr]_u_ [23].

### 4.4. Estimated Glomerular Filtration Rates (eGFR)

The glomerular filtration rate was estimated with equations from the Chronic Kidney Disease Epidemiology Collaboration (CKD-EPI) [27]. CKD stages 1, 2, 3, 4, and 5 corresponded to eGFR of 90–119, 60–89, 30–59, 15–29 and <15 mL/min/1.73 m^2^, respectively. For dichotomous comparisons, CKD was defined as eGFR < 60 mL/min/1.73 m^2^.

### 4.5. Statistical Analysis

Data were analyzed with SPSS 17.0 (SPSS Inc., Chicago, IL, USA). The Kruskal–Wallis test was used to assess differences in means among three subsets, and the Pearson chi-squared test was used to assess differences in percentages. The one-sample Kolmogorov–Smirnov test was used to identify departures of continuous variables from a normal distribution, and a base-10 or base-2 logarithmic transformation was applied to variables that showed rightward skewing before they were subjected to parametric statistical analysis. Throughout the text, base-10 and base-2 logarithms are denoted as log(*x*) and log_2_(*x*), respectively.

In our regression analyses (Table 2 and Table 3), independent variables included log(E_Cd_/E_cr_), log(E_Cd_/C_cr_), log(E_NAG_/E_cr_), log(E_NAG_/C_cr_), age, diabetes, gender, hypertension, and smoking. For each independent variable (*x*), we used multivariable logistic regression analyses to ascertain the prevalence odds ratio (POR) for CKD and the corresponding β-coefficient (Table 2). The β-coefficient—i.e., the slope of a line relating the natural log of POR to *x*—thus depicted the effect of a one-unit change in *x* on POR while other independent variables remained constant. Log base-2 transformation of E_Cd_/E_cr_, E_NAG_/E_cr_, E_Cd_/C_cr_, and E_NAG_/C_cr_ permitted estimation of the factor by which POR increased as each ratio was doubled. We employed two models in each logistic regression analysis: model 1 incorporated log_2_(E_Cd_/E_cr_) and log_2_(E_NAG_/E_cr_); model 2 incorporated log_2_(E_Cd_/C_cr_) and log_2_(E_NAG_/C_cr_).

We performed multivariable linear regression analyses in three subsets of the study sample (Table 3). The subsets were defined by gradations of log[(E_Cd_/E_cr_) × 10^3^] (model 1) or log[(E_Cd_/C_cr_) × 10^5^] (model 2). We examined associations of eGFR with log_10_(E_Cd_/E_cr_), log_10_(E_NAG_/E_cr_), log_10_(E_Cd_/C_cr_), log_10_(E_NAG_/C_cr_), and the aforementioned demographic variables. For each model, an adjusted coefficient of determination (R^2^) and standardized β were obtained to indicate, respectively, the total variation in eGFR that was explained by all independent variables, and the strength of association between eGFR and an individual independent variable.

Polynomial regression was used to fit lines and curves to the following scatterplots: eGFR against log(E_Cd_/E_cr_), log(E_NAG_/E_cr_), log(E_Cd_/C_cr_), and log(E_NAG_/C_cr_); log(E_NAG_/E_cr_) against log(E_Cd_/E_cr_); and log(E_NAG_/C_cr_) against log(E_Cd_/C_cr_). A linear model, *y = a + bx*, was adopted if the relationship was monotonic. A quadratic model (second-order polynomial), *y = a + b*_1_*x + b*_2_
*x*^2^, was used if there was a significant change in the direction of the slope (*b*_1_ to *b*_2_) for prediction of dependent variable *y*. In both types of equations, *a* represented the *y*-intercept.

Relationships between *x* and *y* were assessed with R^2^ (the coefficient of determination) and with unstandardized and standardized β coefficients. In linear and quadratic models, R^2^ is the fraction of variation in *y* that is explained by variation in *x*. In linear models, the unstandardized β coefficient is the slope of the linear regression, and the standardized β coefficient indicates the strength of the association between *y* and *x* on a uniform scale. A linear regression method was used to perform slope analyses of quadratic curves relating eGFR to log(E_Cd_/E_cr_), log(E_NAG_/E_cr_), log(E_Cd_/C_cr_), and log(E_NAG_/C_cr_).

In Figure 1E,F, Figure 2E,F and Figure 3E,F, a univariate model analysis was used to derive mean eGFR (Figure 1 and Figure 2), mean log(E_NAG_/E_cr_) (Figure 3), and mean log(E_NAG_/C_cr_) (Figure 3) for men and women separately with adjustment for covariates (including age) and interactions among independent variables. The Cd-excretion levels in subsets of Figure 1E,F and Figure 3E,F are identical to those depicted in subsets of Table 3. Raw data for eGFR were employed in Table 1 and Table 2 and in Figure 1A,C and Figure 2A,C. In all analyses, two-sided *p*-Values ≤ 0.05 were assumed to indicate statistical significance.

## 5. Conclusions

Excretion rates of Cd and NAG elucidate reductions in GFR that result from renal accumulation of Cd. The conventional method for expressing these excretion rates, normalization of urine concentrations to [cr]_u_, incorporates conceptual flaws that are eliminated if the rates are normalized to C_cr_. In a large and diverse sample of Thai subjects, the alternate approach strengthened all identifiable relationships between eGFR and conventionally quantified urine components, and exposed additional relationships that were obscured by the conventional method. Normalization to C_cr_ should replace normalization to [cr]_u_ in studies that relate urine composition to Cd-induced diminution of eGFR.

## Figures and Tables

**Figure 1 ijms-22-01762-f001:**
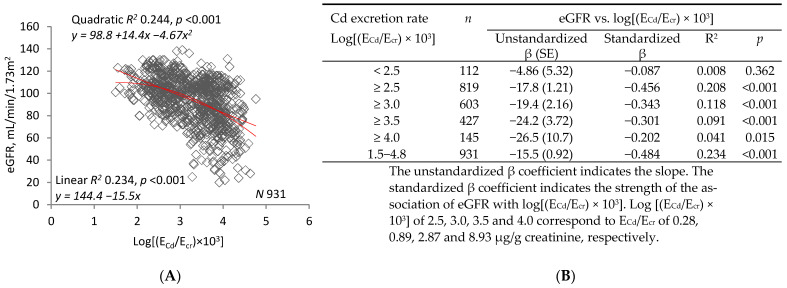
Inverse relationships of eGFR to parameters of Cd excretion. Parameter abbreviations are summarized at the end of the text. Scatterplots in graphs (**A**,**C**) relate eGFR to log[(E_Cd_/E_cr_) × 10^3^] and log[(E_Cd_/C_cr_) × 10^5^] in all subjects. Linear and quadratic equations, their respective coefficients of determination (R^2^), and associated *p*-values are provided. In Table (**B**), linear relationships of eGFR to log[(E_Cd_/E_cr_) × 10^3^] < 2.5, ≥2.5, ≥3.0, ≥3.5 and ≥4.0 are characterized with respective values of subject number (*n*), unstandardized and standardized β, R^2^, and *p*. In Table (**D**, linear relationships of eGFR to log[(E_Cd_/C_cr_) × 10^5^] of <2.5, ≥2.5, ≥3.0, ≥3.5 and ≥4.0 are characterized with respective values of subject number (*n*), unstandardized and standardized β, R^2^, and *p*. In graphs (**E**,**F**), bars represent mean eGFR in women and men grouped by ranges of log[(E_Cd_/E_cr_) × 10^3^] or log[(E_Cd_/C_cr_) × 10^5^]. The subsets thus created are identical to those constituting Cd excretion levels 1, 2, and 3 in Table 3. The letters a and b identify reference values in women and men, respectively, at the lowest rates of Cd excretion. Where appropriate, statistical comparisons are made within each gender between mean eGFR in bars a and b and mean eGFR at higher rates of Cd excretion. Geometric mean (GM) values (standard deviation, SD) of E_Cd_/E_cr_ are 0.15 (0.07) µg/g creatinine at level 1, 2.35 (2.42) µg/g creatinine at level 2, and 14.91(7.57) µg/g creatinine at level 3. GM (SD) of [(E_Cd_/C_cr_) × 100] are 0.15 (0.08) µg/L at level 1, 2.20 (2.25) µg/L at level 2, and 16.01 (11.05) µg/L at level 3.

**Figure 2 ijms-22-01762-f002:**
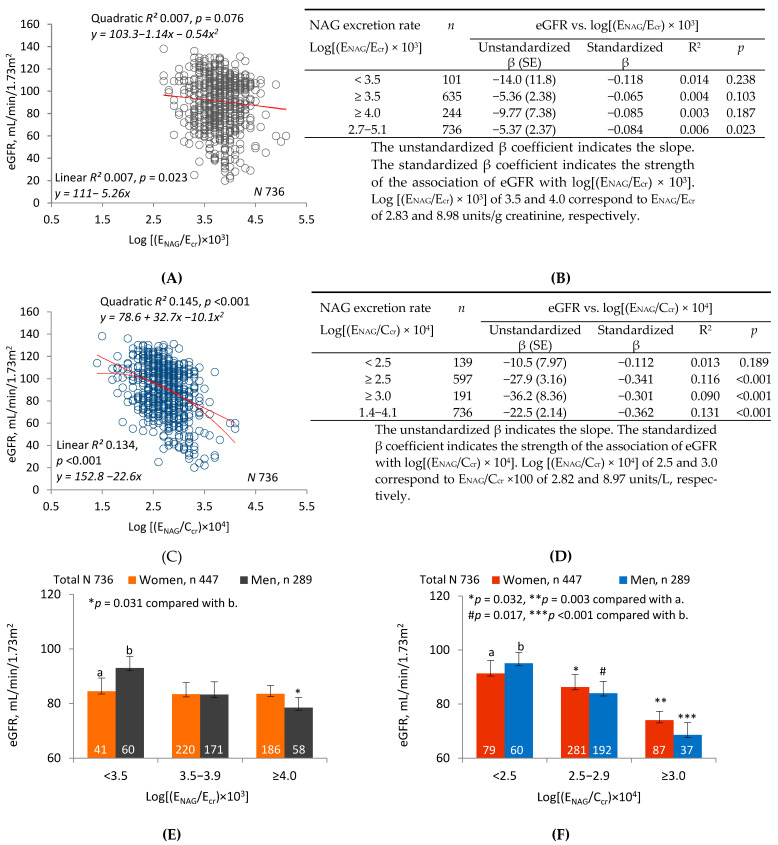
Inverse relationships of eGFR to parameters of NAG excretion. Parameter abbreviations are summarized at the end of the text. Scatterplots in graphs (**A**,**C**) relate eGFR to log[(E_NAG_/E_cr_) × 10^3^] and log[(E_NAG_/C_cr_) × 10^4^] in all subjects. Linear and quadratic equations, their respective coefficients of determination (R^2^), and associated *p*-values are provided. In Table (**B**), linear relationships of eGFR to log [(E_NAG_)/E_cr_) × 10^3^] < 3.5, ≥3.5 and ≥4.0 are characterized with respective values of subject number (*n*), unstandardized and standardized β, R^2^, and *p*. In Table (**D**), linear relationships of eGFR to log[(E_NAG_/C_cr_) × 10^4^] < 2.5, ≥2.5 and ≥3.0 are characterized with respective values of *n*, unstandardized and standardized β, R^2^*,* and *p*. In graphs (**E**,**F**), the bars represent the mean eGFR in women and men grouped by ranges of log[(E_NAG_/E_cr_) × 10^3^] or log[(E_NAG_/C_cr_) × 10^4^] (NAG excretion levels 1, 2, and 3 from lowest to highest). Numbers of women and men within each NAG excretion level are provided. The letters a and b identify reference values in women and men, respectively, at the lowest rates of Cd excretion. Where appropriate, statistical comparisons are made within each gender between mean eGFR in bars a and b and mean eGFR at higher rates of Cd excretion. The geometric mean (GM) (SD) of E_NAG_/E_cr_ in groups 1, 2 and 3 is 1.75 (0.65), 5.36 (1.68) and 14.50 (11.98) units/g creatinine, respectively. The GM (SD) of [(E_NAG_/C_cr_) × 100] in groups 1, 2 and 3 is 1.73 (0.66), 5.56 (2.25) and 17.52 (17.16) units/L, respectively.

**Figure 3 ijms-22-01762-f003:**
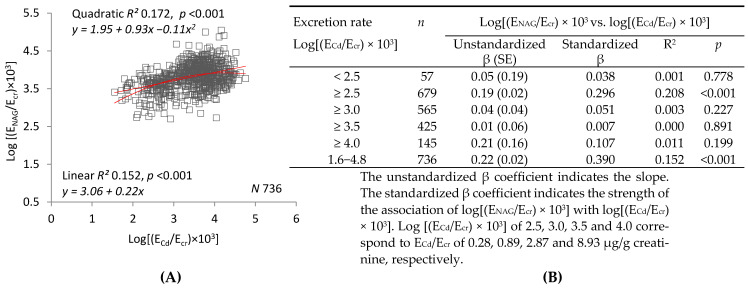
Direct relationships of parameters of NAG excretion to parameters of Cd excretion. Parameter abbreviations are summarized at the end of the text. Scatterplots in graphs (**A**,**C**) relate log[(E_NAG_/E_cr_) × 10^3^] to log[(E_Cd_/E_cr_) × 10^3^], and log[(E_NAG_/C_cr_) × 10^4^] to log[(E_Cd_/C_cr_) × 10^5^] in all subjects. Linear and quadratic equations, their respective coefficients of determination (R^2^), and associated *p*-values are provided. In Table (**B**), linear relationships of log[(E_NAG_/E_cr_) × 10^3^] to log [(E_Cd_/E_cr_) × 10^3^] < 2.5, ≥2.5, ≥3.0, ≥3.5 and ≥4.0 are characterized with respective values of subject number (*n*), unstandardized and standardized β, R^2^, and *p*. In Table (**D**), linear relationships of log[(E_NAG_/C_cr_) × 10^4^] to log [(E_Cd_/C_cr_) × 10^5^] < 2.5, ≥2.5, ≥3.0, ≥3.5 and ≥4.0 are characterized with respective values of subject number (*n*), unstandardized and standardized β, R^2^, and *p*. In graphs (**E**,**F**), the bars represent mean log[(E_NAG_/E_cr_) × 10^3^] or mean log[(E_NAG_/C_cr_) × 10^5^] in women and men grouped by ranges of log[(E_Cd_/E_cr_) × 10^3^] or log[(E_Cd_/C_cr_) × 10^5^]. The subsets thus created are identical to those constituting Cd excretion levels in Table 3 and in Figure 1E,F. Numbers of subjects within each Cd excretion level and corresponding values of GM (SD) of E_Cd_/E_cr_ and [(E_Cd_/C_cr_) × 100] are as described in the legend for Figure 1. The letters a and b identify reference values in women and men, respectively, at the lowest rates of Cd excretion. Where appropriate, statistical comparisons are made within each gender between mean eGFR in bars a and b and mean eGFR at higher rates of Cd excretion.

**Table 1 ijms-22-01762-t001:** Study subjects grouped by urinary cadmium excretion rates normalized to creatinine clearance.

Parameters/Factors	All Subjects	Log[(E_Cd_/C_cr_) × 10^5^]		
	*n* = 931	<2.5, *n* = 146	2.5−3.9, *n* = 654	≥4.0, *n* = 131
Age (years)	44.4 ± 12.5	31.9 ± 9.1	44.8 ±10.9	56.3 ± 10.8 *
Age range	16−87	16−53	18−87	36−83
eGFR (mL/min/1.73 m^2^) ^a^	93.8 ± 20.8	109.3 ± 12.0	95.3 ± 17.7	69.1 ± 21.8 *
eGFR range	20−139	78−130	25−139	20−112
eGFR <60 mL/min/1.73 m^2^ (%)	7.1	0	3.5	32.8 ^†^
Women (%)	58.5	54.1	60.4	54.2
Smoking (%)	39.2	19.2	29.8	58.8 ^†^
Hypertension (%)	25.5	8.2	28.8	26.0 ^†^
Diabetes mellitus (%)	1.2	0	0.6	5.3 ^†^
Serum creatinine, mg/dL	0.85 ± 0.26	0.80 ± 0.15	0.82 ± 0.20	1.08 ± 0.42 *
Urine creatinine, mg/dL	81.1 ± 73.7	53.9 ± 63.0	86.7 ± 76.0	91.2 ± 63.9 *
Urine Cd, μg/L	1.52 ± 8.32	0.11 ± 0.17	1.77 ± 3.98	13.57 ± 14.62 *
Urine NAG, units/L	5.48 ± 9.95	1.73 ± 1.67	6.14 ± 10.70	7.19 ± 7.00 *
Normalized to E_cr_ as E_x_/E_cr_ ^b^				
E_Cd_/E_cr_, µg/g creatinine	1.85 ± 6.23	0.19 ± 0.10	2.04 ± 2.81	14.88 ± 8.05 *
E_NAG_/E_cr_, units/g creatinine	6.40 ± 8.94	3.27 ± 2.39	6.74 ± 7.63	7.88 ± 13.97 *
Normalized to C_cr_ as E_x_/C_cr_ ^c^				
E_Cd_/C_cr_ × 100, µg/L	1.57 ± 7.23	0.15 ± 0.07	1.67 ± 2.24	16.01 ± 11.05 *
E_NAG_/C_cr_ × 100, units/L	5.42 ± 9.39	2.47 ± 1.93	5.47 ± 7.62	8.48 ± 15.12 *

*n* = number of subjects; ^a^ eGFR = estimated glomerular filtration rate, determined with Chronic Kidney Disease Epidemiology Collaboration (CKD−EPI) equations [24]; ^b^E_x_/E_cr_ = [*x*]_u_/[cr]_u_; ^c^E*_x_*/C_cr_ = [*x*]_u_[cr]_p_/[cr]_u_, where *x* = Cd or NAG [23]. Data for age and eGFR are arithmetic means ± standard deviation (SD). Data for all other continuous variables are geometric means ± SD. Data for urine NAG are from 736 subjects; data for all other variables are from 931 subjects. ^†^ Significant % differences among three groups (*p* < 0.001, Pearson Chi-Square test). * Significant mean differences among three groups (*p* < 0.001, Kruskal–Wallis test).

**Table 2 ijms-22-01762-t002:** Relationships of prevalence odds ratios for chronic kidney disease to demographic factors and parameters of Cd and NAG excretion.

Independent Variables/Factors	eGFR Levels <60 mL/min/1.73 m^2^				
	β Coefficients	POR	95% CI		*p*
	(SE)		Lower	Upper	Value
*Model 1, n = 736*					
Age (years)	0.143 (0.018)	1.153	1.113	1.195	<0.001
Log_2_ [(E_Cd_/E_cr_) × 10^3^], µg/g creatinine	0.493 (0.137)	1.637	1.252	2.141	<0.001
Log_2_ [(E_NAG_/E_cr_) × 10^3^], units/g creatinine	0.435 (0.172)	1.544	1.103	2.163	0.011
Diabetes	0.705 (0.884)	2.023	0.358	11.434	0.425
Gender (women)	−0.014 (0.374)	0.986	0.474	2.054	0.971
Hypertension	0.644 (0.341)	1.903	0.976	3.711	0.059
Smoking	−0.211(0.372)	0.810	0.391	1.679	0.571
Adjusted R^2^	0.499	−	−	−	<0.001
*Model 2, n = 736*					
Age (years)	0.149 (0.021)	1.160	1.113	1.210	<0.001
Log_2_ [(E_Cd_/C_cr_) × 10^5^], µg/L	0.843 (0.161)	2.324	1.695	3.187	<0.001
Log_2_ [(E_NAG_/C_cr_) × 10^4^], units/L	0.985 (0.192)	2.678	1.837	3.905	<0.001
Diabetes	0.100 (0.961)	1.105	0.168	7.264	0.917
Gender (women)	0.459 (0.429)	1.582	0.682	3.669	0.285
Hypertension	0.667 (0.395)	1.949	0.898	4.229	0.091
Smoking	−0.374 (0.414)	0.688	0.305	1.549	0.366
Adjusted R^2^	0.544	−	−	−	<0.001

POR = Prevalence Odds Ratio; S.E. = Standard error of mean; Data were generated from logistic regression analyses relating POR for CKD to independent variables. Independent variables are listed in the first column. CKD was defined as eGFR <60 mL/min/1.73m^2^. *p*-Values < 0.05 indicate a statistically significant increase in POR for CKD. Log_2_[(E_Cd_/E_cr_) × 10^3^] and log_2_[(E_NAG_/E_cr_) × 10^3^] were incorporated into model 1; log_2_[(E_Cd_/C_cr_) × 10^5^] and log_2_[(E_NAG_/C_cr_) × 10^4^] were incorporated into model 2. Other independent variables in models 1 and 2 were identical. β coefficients indicate the size of the effect of an independent variable on POR for CKD. Adjusted R^2^ values were obtained by univariate analyses that incorporated eGFR as a continuous variable. Independent variables were identical to those used in the logistic regression analyses.

**Table 3 ijms-22-01762-t003:** Relationships of eGFR to demographic factors and parameters of Cd and NAG excretion.

Independent Variables/Factors	eGFR, mL/min/1.73 m^2^					
	Cd Excretion Level 1		Cd Excretion Level 2		Cd Excretion Level 3	
	β	*p*	β	*p*	β	*p*
*Model 1, n = 736*						
Age (years)	−0.527	<0.001	−0.624	<0.001	−0.489	<0.001
Log_10_[(E_Cd_/E_cr_) × 10^3^],µg/g creatinine	−0.077	0.422	−0.088	0.023	−0.138	0.061
Log_10_[(E_NAG_/E_cr_) × 10^3^], unit/g creatinine	0.368	<0.001	0.068	0.060	0.056	0.456
Gender (women)	−0.261	0.026	−0.053	0.166	−0.030	0.715
Smoking	−0.256	0.025	0.029	0.455	−0.069	0.392
Hypertension	0.266	0.013	0.049	0.139	0.127	0.091
Diabetes	−	−	0.041	0.201	0.135	0.069
Adjusted R^2^	0.531	<0.001	0.454	<0.001	0.290	<0.001
*Model 2, n = 736*						
Age (years)	−0.640	<0.001	−0.548	<0.001	−0.483	0.001
Log_10_[(E_Cd_/C_cr_) × 10^5^], µg/L	−0.073	0.414	−0.128	0.001	−0.281	0.001
Log_10_[(E_NAG_/C_cr_) × 10^4^], units/L	0.175	0.049	−0.095	0.010	−0.228	0.002
Gender (women)	−0.276	0.010	−0.091	0.023	0.068	0.366
Smoking	−0.109	0.276	−0.040	0.335	0.002	0.980
Hypertension	0.168	0.075	0.029	0.420	0.108	0.125
Diabetes	−	−	0.053	0.125	0.026	0.706
Adjusted *R*^2^	0.452	<0.001	0.383	<0.001	0.436	<0.001

Data were derived from two linear regression models relating eGFR, a continuous dependent variable, to seven independent variables (first column). We incorporated log_10_[(E_Cd_/E_cr_) × 10^3^] and log_10_[(E_NAG_/E_cr_) × 10^3^] into model 1 and log_10_[(E_Cd_/C_cr_) × 10^5^] and log_10_[(E_NAG_/C_cr_) × 10^4^] into model 2. Units of E_Cd_/E_cr_ and E_NAG_/E_cr_ are µg/g cr and units/g cr, respectively. Units of E_Cd_/C_cr_ and E_NAG_/C_cr_ are µg/L of filtrate and units/L of filtrate, respectively. Cd excretion level 1 implies log_10_[(E_Cd_/E_cr_) × 10^3^] and log_10_[(E_Cd_/C_cr_) × 10^5^] < 2.5. Cd excretion level 2 implies log_10_[(E_Cd_/E_cr_) × 10^3^] and log_10_[(E_Cd_/C_cr_) × 10^5^] = 2.5–3.9. Cd excretion level 3 implies log_10_[(E_Cd_/E_cr_) × 10^3^] and log_10_[(E_Cd_/C_cr_) × 10^5^] ≥ 4.0. Standardized regression coefficients (β) indicate strength of associations between eGFR and independent variables. *p* ≤ 0.05 identifies statistical significance. Adjusted R^2^ values indicate the total variation in eGFR that was explained by all independent variables incorporated into each model. For model 1, the values for geometric mean (GM) (SD) of E_Cd_/E_cr_ within Cd excretion levels 1, 2 and 3 were 0.15 (0.07), 2.35 (2.42) and 14.9 (7.57) µg/g creatinine. The corresponding numbers of subjects in the three Cd excretion levels were 57, 534 and 145, respectively. For model 2, the values for GM (SD) of [E_Cd_/C_cr_ × 100] (SD) within Cd excretion levels 1, 2 and 3 were 0.15 (0.08), 2.20 (2.25) and 16.01 (11.05) µg/L of filtrate. The corresponding numbers of subjects in the three Cd excretion levels were 82, 523 and 131, respectively.

## Data Availability

Data are contained within the article.

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
