# Peer review of "The Effect of Cadmium on GFR Is Clarified by Normalization of Excretion Rates to Creatinine Clearance"

_ijms, 2021, doi:10.3390/ijms22041762_

Round 1

Reviewer 1 Report

The manuscript entitled, “The effect of cadmium on GFR is clarified by normalization of excretion rates to creatinine clearance” is a very well written and organized manuscript attempting to show how creatinine clearance expressed as a coefficient should replace normalization of nephrotoxicity markers to urinary creatinine when assessing eGFR. However, major limitations and explanation provisions exist which should be addressed before consideration for publication:

Major:

  1. In reference to Figures 1, 2, and 3 E & F, the significant relationship of eGFR to age, were discussed; however, there is concern about normalization of this effect to the significant differences observed between women and men. Could the significant differences in eGFR observed between women and men as cadmium excretion levels increase be convoluted by age in these cohorts?
  2. Overall conclusions of manuscript seem to be aligned with much of what is already known in the literature in regards to cadmium-induced nephrotoxicity. There is concern about what new information this adds to the field.
  3. It seems that many of the calculations utilized to compare eGFR to cadmium levels assume a consistent proximal tubule number across the study population; however, this is highly variable. How can this be explained or normalized?

Author Response

RESPONSES TO REVIEWER 1

Point 1:  In reference to Figures 1, 2, and 3 E & F, the significant relationship of eGFR to age, were discussed; however, there is concern about normalization of this effect to the significant differences observed between women and men. Could the significant differences in eGFR observed between women and men as cadmium excretion levels increase be convoluted by age in these cohorts?

Response: 

We thank the reviewer for raising this question.  Although we stated in the original submission that the analyses in question had been adjusted for covariates and interactions, we did not single out age for the emphasis that it deserves.  In our revised Materials and Methods section (lines 476-477), we have mentioned age specifically as one of the covariates.  In the Results section, we have reiterated in descriptions of Figures 1-3 that analyses were adjusted for covariates and interactions (lines 167, 171, 198, 201, 228, 231-232).

Although the reviewer states otherwise, we did not compare results in women to results in men in these figures.  We compared results of gradations of Cd exposure within a single gender.  The only significant effect of gender on any analysis in the present paper is documented in Table 3; in the model incorporating normalization of excretion rates to Ccr, female gender increased the steepness of the slope relating eGFR to low or moderate Cd exposure.  The increased susceptibility of women to Cd nephrotoxicity is generally attributed to the relationship of low iron stores to enhanced absorption of ingested Cd (For a discussion, see Satarug et al, Stresses 2021, 1:3-15).  Because this issue is far removed from the central message of the present paper, we have not addressed it in the text.

Point 2:  Overall conclusions of manuscript seem to be aligned with much of what is already known in the literature in regards to cadmium-induced nephrotoxicity. There is concern about what new information this adds to the field.

Response: 

We did not intend that the present paper would add new information about the role of Cd accumulation in tubular cell injury and deterioration of GFR.  We have recently examined these issues in depth (Satarug et al, Toxics, 2019, 7, 55; Toxics 2020; 8, 86; Satarug and Phelps, Metal Toxicology Handbook, 2021, pp. 219-274).  The central message of the current submission is summarized in the title: “The effect of cadmium on GFR is clarified by normalization of excretion rates to creatinine clearance.”  In the original version, we stated a hypothesis to this effect in the Abstract, but not in the Introduction.  We have corrected this omission in the revision (lines 68-69).

For the past 30-40 years, investigators of Cd nephropathy have consistently normalized relevant excretion rates, e.g., of Cd and N-acetylglucosaminidase (NAG), to creatinine excretion (Ex/Ecr).  After simplification, Ex/Ccr becomes [x]u/[cr]u, which is usually reported.  In two recent papers, we abandoned the conventional practice and instead normalized Ex to creatinine clearance (Ccr) (Satarug et al, Clin Kidney J 2018, 12, 468-475; Toxics, 2019, 7, 55).  We have previously offered theoretical arguments for making this change (Satarug et al, Toxics, 2019, 7, 55; Toxics 2020, 8, 86), and we summarize them in the Introduction of the present submission (lines 52-67).  In brief, normalization of Ex to Ccr has two salutary effects:  it nullifies the effect of muscle mass on Ecr and [cr]u, and it corrects for the effect of nephron number on excretion of substances emanating from tubular cells. 

In the present paper, we hypothesized that normalization of ECd and ENAG to Ccr would clarify relationships of Cd-induced cellular damage to eGFR.  We examined the hypothesis with a multivariable logistic regression analysis (Table 2), a multivariable linear regression analysis (Table 3), and three univariate linear and quadratic regression analyses (Figures 1-3).  In comparison to the conventional method, normalization of ECd and ENAG to Ccr consistently strengthened relationships between these excretion rates and either eGFR (Table 3; Figures 1-3) or the odds of CKD (Table 2), and also exposed associations that would otherwise have been missed (Table 3).  Because our results are densely presented, we have attempted to help the reader with summaries in the Discussion that enumerate our findings and emphasize their originality and utility (lines 276-286 and 316-320).  These summaries were included in the original submission.

We introduced normalization of excretion rates to Ccr to the field of Cd nephropathy, and three years later, we are still the only investigators who have employed this approach in published research.  Consequently, we are also the first group to compare results of normalizing excretion rates to Ecr or CcrWith respect to our hypothesis, all of the information in the present paper is new.  Because the results are unequivocal, they suggest that general adoption of our method would improve the quality of research in Cd nephropathy.  

Point 3:  It seems that many of the calculations utilized to compare eGFR to cadmium levels assume a consistent proximal tubule number across the study population; however, this is highly variable. How can this be explained or normalized?

Response:

We do not understand how the reviewer came to this interpretation.  We subscribe to the widely shared view that in CKD, GFR and (more approximately) Ccr are functions of the number of intact nephrons.  A reduction in GFR or Ccr implies a proportional reduction in nephron number.  The unit of Ex/Ccr is amount of x excreted per volume of filtrate; because the number of nephrons determines Ccr, Ex/Ccr is proportional to the amount of x excreted per intact nephron

Since Ccr varies among subjects, nephron number does as well.  The act of normalizing Ex to Ccr is an acknowledgement of this fact, and if substance x emanates from tubular cells, this normalization nullifies the effect of nephron number on Ex.  This is one of the two theoretical reasons to normalize Ex to Ccr.  We do not assume a consistent number of proximal tubules.

Reviewer 2 Report

The manuscript "The Effect of Cadmium on GFR Is Clarified by Normalization of Excretion Rates to Creatinine Clearance" is very, very good paper. Authors used regression analysis to presentation of relationhips of eGFR to demographic factors and parameters of Cd and NAG excretion as well as to inverse relationships of eGFR to parameters of Cd excretion.
Table 2: Lack of coefficients of determination for obtained models.
In Figure 1: (B) and (D) are Tables not Figures.
In References lack of new citations.
In my opinion paper needs minlr revision.

Author Response

RESPONSES TO REVIEWER 2

Point 1:  In Table 2, lack of coefficients of determination for obtained models.

Response:

We have supplied adjusted R2 values for both models in Table 2, and have added relevant commentary at the end of the table footnotes.

Point 2:  In Figure 1, (B) and (D) are tables, not figures.

Response:

We have labeled (B) and (D) as tables in the Results section (line 161) and in the legend to Figure 1. 

Point 3:  References lack new citations.

Response:

We have added a recent review of ours that is particularly germane to the point about nephron number raised by Reviewer 1 (reference 3), and three more recent papers relating ENAG to ECd (references 9-11).

Round 2

Reviewer 1 Report

I am fine with the authors' responses to my reviews, and would recommend moving forward to Accept.